# Elevated blood pressure in high-fat diet-exposed low birthweight rat offspring is most likely caused by elevated glucocorticoid levels due to abnormal pituitary negative feedback

**Takahiro Nemoto** [1] *, **Takashi Nakakura** [2], **Yoshihiko Kakinuma** [1]

**1** Department of Bioregulatory Science (Physiology), Nippon Medical School, Tokyo, Japan, **2** Department of Anatomy, Graduate School of Medicine, Teikyo University, Tokyo, Japan

* taknemo@nms.ac.jp

**Data Availability Statement:** All relevant data (mean, SD, and SE with all raw data) are shown in S2 Table. Raw western blot data is shown in the

## Abstract

Being delivered as a low birthweight (LBW) infant is a risk factor for elevated blood pressure and future problems with cardiovascular and cerebellar diseases. Although premature babies are reported to have low numbers of nephrons, some unclear questions remain about the mechanisms underlying elevated blood pressure in full-term LBW infants. We previously reported that glucocorticoids increased miR-449a expression, and increased miR-449a expression suppressed Crhr1 expression and caused negative glucocorticoid feedback. Therefore, we conducted this study to clarify the involvement of pituitary miR-449a in the increase in blood pressure caused by higher glucocorticoids in LBW rats. We generated a fetal low-carbohydrate and calorie-restricted model rat (60% of standard chow), and some individuals showed postnatal growth failure caused by growth hormone receptor expression. Using this model, we examined how a high-fat diet (lard-based 45kcal% fat)-induced mismatch between prenatal and postnatal environments could elevate blood pressure after growth. Although LBW rats fed standard chow had slightly higher blood pressure than control rats, their blood pressure was significantly higher than controls when exposed to a high-fat diet. Observation of glomeruli subjected to periodic acid methenamine silver (PAM) staining showed no difference in number or size. Aortic and cardiac angiotensin II receptor expression was altered with compensatory responses. Blood aldosterone levels were not different between control and LBW rats, but blood corticosterone levels were significantly higher in the latter with high-fat diet exposure. Administration of metyrapone, a steroid synthesis inhibitor, reduced blood pressure to levels comparable to controls. We showed that high-fat diet exposure causes impairment of the pituitary glucocorticoid negative feedback via miR-449a. These results clarify that LBW rats have increased blood pressure due to high glucocorticoid levels when they are exposed to a high-fat diet. These findings suggest a new therapeutic target for hypertension of LBW individuals.

Supporting Information figure. The figure of tissue staining is the original image of the micrograph.

**Funding:** This study was supported in part by JSPS KAKENHI Grant Number 17K10195 to T. Nemoto.

**Competing interests:** The authors have declared that no competing interests exist.

**Abbreviations:** AT1, angiotensin II type 1; AT2, angiotensin II type 2; CVD, cardiovascular disease; CRF, corticotropin-releasing factor; DOHaD, developmental origins of health and disease; HFD, high-fat diet; HPA, hypothalamic-pituitary-adrenal; IUGR, intrauterine growth restriction; LBW, low birthweight; LC, low-carbohydrate and calorie-restricted diet; PAM, periodic acid methenamine silver; SGA, small for gestational age; SMA, α-smooth muscle.

## Introduction

Hypertension is a multifactorial disease caused by the interaction of genes and environment, and is a major factor in the future development of cardiovascular disease (CVD) [1, 2]. Risk factors for developing hypertension vary, but one is considered to be low birthweight (LBW). A systematic review by Huxley *et al.* found that birthweight and blood pressure are negatively correlated [3]. While studying the morbidity and mortality of a large cohort in the general population, Barker and colleagues found a strong association between birthweight and susceptibility to both adult onset hypertension and CVD [4]. Many other epidemiological and experimental studies have strongly supported the Barker's findings [5–7]. There is also a link between LBW and an increased risk of death of individuals from CVD who are born small and more likely to have left ventricular hypertrophy [8] and coronary heart disease [9]. Furthermore, the developmental origins of health and disease (DOHaD) theory has evolved into the idea that mismatches between the acquired constitution in utero and the postnatal growth environment create a risk of developing noncommunicable diseases [10–14].

In Japan, an increase in the rate of LBW infants has been reported to be remarkably elevated for over a decade, and it is an urgent task to control the future development of non-communicable chronic diseases in these children [15, 16]. However, verification of the mechanisms underlying the adult onset of hypertension caused by LBW in humans is time consuming and has many ethical problems. Therefore, we generated a model rat that was fed a low-carbohydrate and calorie-restricted diet during pregnancy. Although the rats did not deliver prematurely, the offspring weighed less and some of them failed to catch up during postnatal growth. In addition, we found that when they were exposed to restraint stress, blood corticosterone levels were further elevated due to a dysfunctional glucocorticoid-mediated feedback system in the pituitary gland [17, 18].

The mechanisms that mediate fetal programming of hypertension have been extensively studied and reviewed in three major organs or systems [19–21]: the kidney (decreased nephron number, activation of the renin-angiotensin system, or increased renal sympathetic nerve activity), the vasculature (alterations in structure or impaired vasodilation), and the neuroendocrine system [hypothalamic-pituitary-adrenal (HPA) upregulation or altered adaptation to stress]. The first causative organ responsible for hypertension is the kidney. Births of a low weight or being premature are associated with an increased risk of hypertension, proteinuria, and kidney diseases that develop later in life. Adverse events in the uterus can affect fetal kidney development and reduce the final number of nephrons [22]. The octapeptide angiotensin II, which is generated from angiotensin I by the angiotensin-converting enzyme, modulates blood pressure, water and sodium homeostasis, neuronal function, and other neurohumoral systems. The effects of angiotensin II are mediated by two receptors, referred to as the angiotensin II type-1 (AT1) and type-2 (AT2) receptor subtypes. The AT1 and AT2 are both G protein-coupled receptors, but AT2 shares only 34% identity with AT1 [23, 24]. AT1-receptor are expressed in vascular smooth muscle cells, heart, lung, brain, liver, kidney, adrenal glands, and others and its activation causes vasoconstriction, sodium and water retention, neurohumoral activation, and increased vascular reactive oxygen species (ROS) production. AT2 expression is found in the kidney and nervous and cardiovascular systems of adult rats [25, 26]. It has already been summarized that AT2 inhibits the AT1-mediated increase in inositol triphosphate and shows vasodilator, antifibrotic, antiproliferative, and anti-inflammatory effects [27–30].

The second causative organ for hypertension is the vasculature. Well-established animal models indicate that environmentally induced intrauterine growth restriction (IUGR); by diet, diabetes, hormonal exposure, or hypoxia, increases the risk of development of various diseases

in target organs later in life [31]. This suggests that a series of perturbations in fetal and postnatal growth may influence developmental programming and produce abnormal phenotypes. The endothelium, composed of endothelial cells, may be an important player for modulating long-term remodeling and the elastic properties of arterial walls.

The third cause is the neuroendocrine system. We previously reported that LBW rats maintain elevated blood corticosterone under restraint stress [17, 18]. Therefore, we investigated whether a high-fat diet, which is a chronic metabolic stress, could increase blood corticosterone levels in LBW rats. We have already shown that pituitary miR-449a is involved in the negative feedback of glucocorticoids [32], and restraint stress-induced LBW rats have impaired induction of miR-449a expression in the anterior pituitary gland.

Therefore, the purpose of the present study was to clarify whether LBW caused by low carbohydrate and calorie restriction in fetuses increases blood pressure after growth, and if a mismatch between the acquired constitution in the fetal age and the postnatal growth environment further can increases blood pressure, and which organs are involved. To this end, we investigated the morphology of the kidneys of LBW rats and examined the number and size of glomeruli, and the expression levels of AT1 and AT2 in the abdominal aorta (one of the common sites of arteriosclerosis in humans) and heart. We also examined the relationship between blood adrenal steroid levels and blood pressure, and investigated whether pituitary glucocorticoid feedback in LBW rats is affected by high-fat diet exposure. Finally, to determine whether elevated corticosterone is responsible for increasing blood pressure, we examined whether administration of metyrapone can normalize blood pressure. Thus, we have investigated the mechanism by which hypertension occurs in LBW rats exposed to a high-fat diet.

## Methods

### Animals

Wistar rats were maintained at 23 ± 2˚C with a 12:12-h light-dark cycle (lights on at 0800 h, off at 2000 h). They were allowed *ad libitum* access to laboratory chow and sterile water. All experimental procedures were reviewed and approved by the Laboratory Animals Ethics Review Committee of Nippon Medical School (#27–067 and #2020–003). All experiments were performed in accordance with relevant guidelines and regulations [33]. We previously generated fetal low-carbohydrate and calorie-restricted rats [34]. Briefly, twenty proestrous female rats (age, 9 weeks) were mated with normal male rats. Dams were housed individually with free access to water and were divided into two groups: low-carbohydrate and calorie-restricted diet (LC) dams were restricted in their calorie intake to 60% of the control group during the entire gestational period (S1 Table, D08021202, Research Diet Inc., New Brunswick, NJ), while control dams freely accessed food during the period. Twelve to twenty pups were obtained from 10 dams of each group. We excluded rat pups born with a body weight of more than 6.0 g, which is the average-2SD body weight of the offspring of normal dams. No surrogate mother was used, and 10 rat pups were left at random and raised under the birth mother rat. Postnatal mother rats were fed a standard diet *ad libitum*. After weaning, the rats of different litters were mixed. Male rats (n = 46) were divided into two groups at 5 weeks of age, one on a high-fat diet (lard-based 45kcal% fat, D12451, Research Diet) (n = 23) and the other on a standard diet (n = 23) (Fig 1A). At 18 weeks of age, blood pressure was measured, blood was collected, organs were removed from the rats, and gene expressions, hormone levels were analyzed.

### Tissue staining

The abdominal aortas and kidneys of rats were removed and fixed by immersion in 4% paraformaldehyde in 0.1 M phosphate buffer (pH 7.4) for 1 day at 4˚C, dehydrated through a

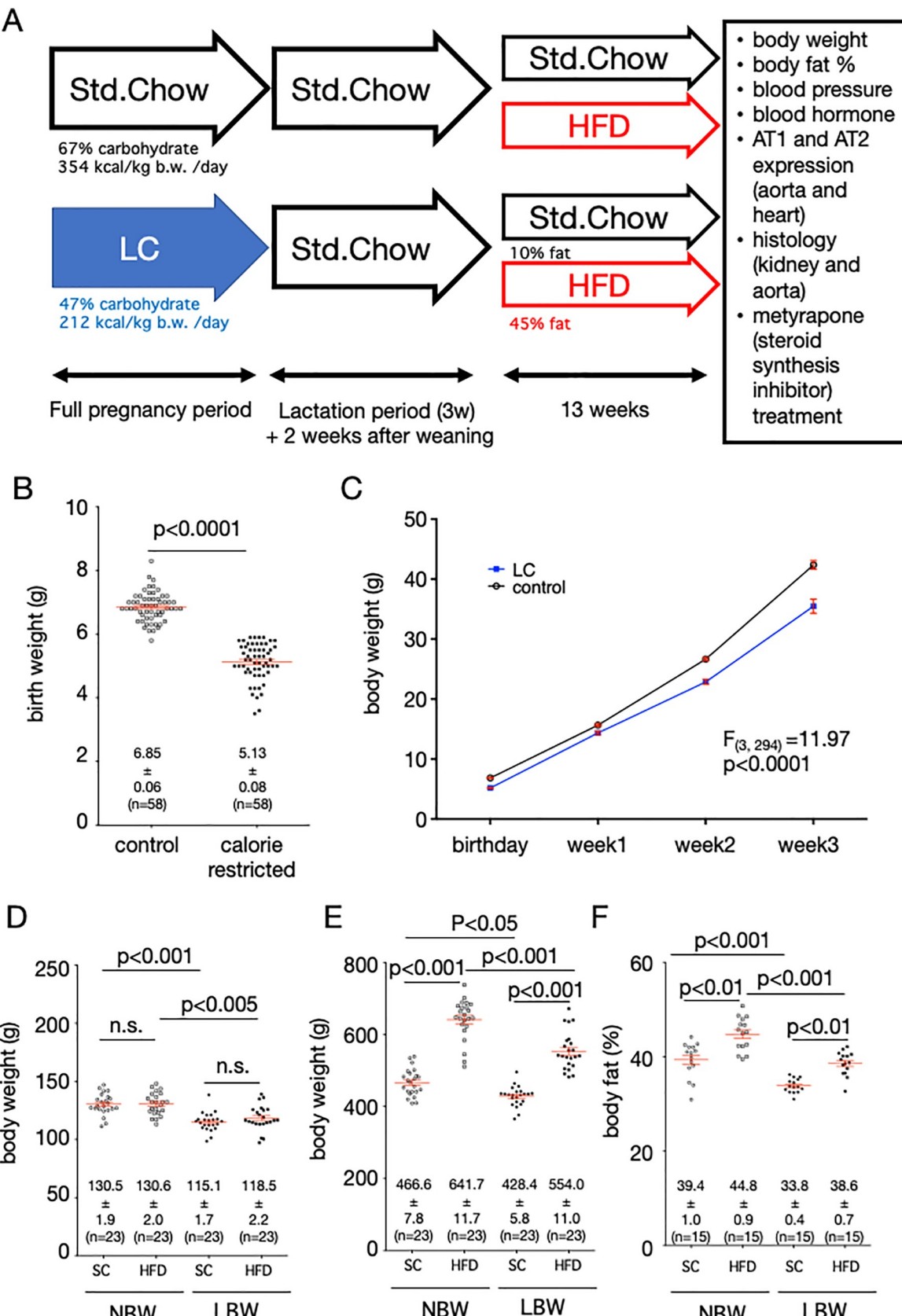

**Fig 1. Experimental schema and rat weight.** A, Schema of experimental design. Std.Chow indicates a standard diet, and LC indicates a low-carbohydrate calorie restricted diet. Below the arrows are shown the composition of the diet and the average calorie intake per day. HFD stands for high-fat diet. The graph of the birth weight of the male rat used in the experiment (B), the alteration of the weight during lactation (C), the weight before (D) and after (E) exposure to the high-fat diet (HFD) or standard chow (SC), and the body fat percentage after exposure to the HFD or SC (F) are shown. Statistical analysis was performed using unpaired T test for B, two-way ANOVA for C, and one-way ANOVA followed by Turkey's *post hoc* test for multiple comparisons for D-F.

graded ethanol series, and embedded in paraffin. The sections were cut with a microtome (SM 2000 R, Leica Biosystems, Wetzlar, Germany) and placed on PLATINUM PRO slides (Matsunami, Osaka, Japan) as previously described [35]. For observation of the renal glomerular basement membranes, kidney sections (1 μm thick) were deparaffinized and stained with periodic acid methenamine silver (PAM). For immunohistochemistry, deparaffinized aorta sections were treated for antigen retrieval by heating in an autoclave in 1 mM EDTA at 121˚C for 5 min, and were then incubated overnight at 25˚C with mouse anti-α-smooth muscle (SMA) (1:200; A5228, Sigma) in phosphate-buffered saline (PBS) containing 1% bovine serum albumin. After washing with PBS, the sections were incubated with Cy3-labeled donkey anti-rabbit IgG, Alexa Fluor 488-labeled donkey anti-mouse IgG (Jackson Immunoresearch, West Grove, PA, USA), and 4',6-diamidino-2-phenylindole (DAPI; Dojindo, Kumamoto, Japan) for 2 h at room temperature. The specimens were examined with a BX53 microscope equipped with a DP80 microscope digital camera and cellSens imaging software (Olympus Optical, Tokyo, Japan).

## Blood pressure and body fat measurements

Blood pressure was measured non-invasively from tail blood volume, flow, and pressure using a volume pressure recording sensor and an occlusion tail cuff (CODA System; Hakubatec Lifescience Solutions, Tokyo, Japan) [36]. As reported previously, this is a highly accurate system that can non-invasively and simultaneously measure systolic and diastolic blood pressure and heart rate [37]. Prior to measurement, rats were placed on a 37˚C heating pad until the tail temperature reached 37˚C. After heating, blood pressure was measured 10 times, and the average value was used. All measurements were performed at the same time (10:00 am to 02:30 pm). The measurement of the rat body fat percentage was performed using ImpediVET (BRC bioresearch center, Nagoya, Japan) under 4% isoflurane anesthesia.

## RNA extraction and real-time RT-PCR

We performed mRNA and miRNA quantification as previously reported [17]. Total RNA was extracted from abdominal aortas, hearts, kidneys, and pituitaries using RNAiso Plus (Takara, Shiga, Japan). The absorbance of each sample at 260 nm and 280 nm was assayed, and RNA purity was judged as the 260/280 nm ratio (The 260 /280 nm ratio of all samples used in this study was higher than 1.7). For miRNA expression analysis, first-strand cDNA was synthesized at 37˚C for 1 h using 500 ng of denatured total RNA and then terminated at 85˚C for 5 min using a Mir-X® miRNA First-Strand Synthesis and SYBR® qRT-PCR kit (Clontech Laboratories Inc., Mountain View, CA). For mRNA expression analyses, first-strand cDNA was generated using 250 ng of denatured total RNA; the reaction mixture was incubated at 37˚C for 15 min, 84˚C for 5 sec, and 4˚C for 5 min using a PrimeScript® RT reagent kit with gDNA Eraser (Takara). PCR was performed by denaturation at 94˚C for 5 sec and annealing-extension at 60˚C for 30 sec for 40 cycles using SYBR premix Ex Taq (Takara) and specific primer sets for rat AT1 (RA061027, Takara), AT2 (RA060278, Takara), or GAPDH (RA015380, Takara). To normalize each sample for RNA content, GAPDH, a housekeeping gene, and U6 small nuclear

RNA (Clontech Laboratories, Inc.) were used for mRNA and miRNA expression analyses, respectively. The $2^{nd}$ derivative method was used as the standard and for calculating $C_t$ values, respectively [38].

## Western blotting

We performed western blotting quantification as previously reported [34]. Protein samples from rat infrarenal aortas and hearts were extracted using complete lysis-M (Roche, Mannheim, Germany). The protein concentrations of lysate samples were determined using the Pierce 660 nm Protein assay (Thermo Scientific, Rockford, IL). Each 5 μg of protein was electrophoresed on a 5–20% gradient SuperSep$^{TM}$ SDS-polyacrylamide gel (FUJIFILM Wako Pure Chemical Corporation, Osaka, Japan) and transferred to a nitrocellulose membrane. The transfer membranes were blocked with 5% skim milk and then incubated with an anti-AT1 antibody (1:1,000, GTX89149, GeneTex, Inc., Irvine, CA) or anti-AT2 antibody (1:1,000, GTX62361, GeneTex) for 1 h at room temperature. The transfer membranes were washed with TBS-T and further incubated with HRP-labeled anti-goat or anti-rabbit IgG (1:2,000, Jackson ImmunoResearch, West Grove, PA) for 1 h at room temperature. The signals were detected using SuperSignal West Dura extended duration substrate (Thermo Scientific). The membranes after detection were stripped of the antibody using Restore plus western blot stripping buffer (Thermo Scientific). Then, signals were detected again using THE$^{TM}$ [HRP]-labeled β-actin antibody (1:1,000, A00730-40, GeneScript, Piscataway, NJ). The expression levels of AT1 or AT2 were quantified by correcting the AT1 or AT2 signal with the β-actin signal.

## Measurement of blood aldosterone, corticosterone and leptin levels

Aldosterone and corticosterone levels were measured using blood plasma from decapitated rats. Aldosterone was measured using a rat aldosterone EIA kit (#501090, Cayman Chemical, Ann Arbor, MI), corticosterone was measured using a rat corticosterone ELISA kit (#501320, Cayman Chemical) and leptin was measured using a Mouse/Rat Leptin Quantikine ELISA kit (R&D systems, Minneapolis, MN).

## Metyrapone administration

Metyrapone inhibits 11-beta-hydroxylase, thereby inhibiting synthesis of cortisol from 11-deoxycortisol and corticosterone from deoxycorticosterone in the adrenal gland. Administration of metyrapone (Sigma-Aldrich, St Louis, MO) to rats was performed according to a previous report [39]. Rats treated with metyrapone (10 mg/100 g body weight, s.c., twice daily at 0900 h and 1800 h for a week).

## Statistical analysis

Unpaired $t$ tests, a one-way analysis of variance (ANOVA) followed by Turkey's *post hoc* test for multiple comparisons, or two-way ANOVA were used for each statistical analysis. Prism 5.0 software (GraphPad Software, Inc., La Jolla, CA) was used for all calculations. Real-time RT-PCR and western blot data are expressed as percents ± SEM with the control set to 100. p <0.05 was considered statistically significant.

## Results

### Body weight and body fat percentage

There were no differences in the gestational times of the offspring or the numbers of births from LC and control dams. The average birthweight of offspring from calorie-restricted

mother rats was 5.13±0.08 g (mean ± SEM, n = 58), which was significantly lower than that of control rats by 6.85±0.06 g (n = 58) (Fig 1B, p<0.0001). The average birthweight of litters by mother is shown in S2 Table. The average bodyweight of LBW rats was low until weaning and did not increase rapidly during lactation (Fig 1C). The average bodyweight of offspring from calorie-restricted mother rats before high-fat diet exposure (5 weeks old) was 118.5±2.2 g (n = 23), which was significantly lower than that of control rats by 130.6±2.0 g (n = 23) (Fig 1D, p<0.005). The bodyweight of LBW rats after high-fat diet exposure (18 weeks old) was 554.0±11.0 g (n = 23), which was significantly lower than that of control rats exposed to high-fat diet, 641.7±11.7 g (n = 23) (Fig 1E, p<0.001). The bodyweight of LBW rats on standard diet was not statistically different from NBW rats on standard diet. Similarly, the body fat percentage of LBW rats after high-fat diet exposure was 38.6±0.7% (n = 15), which was significantly less than the body fat percentage of NBW rats exposed to high-fat diet 44.8±0.9% (n = 15) (Fig 1F, p<0.01).

## Blood pressure

The blood pressure of rats fed the standard diet showed a slight but not significant increase in both systolic (110.9±2.5 vs. 120.2±3.9 mmHg, 18-week-old NBW vs. 18-week-old LBW, respectively, mean ± SEM, n = 15, p>0.05) and diastolic pressure (83.0±1.7 vs. 89.2±2.7 mmHg, NBW vs. LBW, respectively, n = 15, p>0.05) in LBW rats compared with control rats (Fig 2A). When rats were exposed to a high-fat diet, the systolic (133.3±2.2 vs. 151.1±2.5 mmHg, NBW vs. LBW, respectively, n = 15, p<0.001) and diastolic pressures (89.8±2.2 vs. 104.7±1.6 mmHg, NBW vs. LBW, respectively, n = 15, p<0.001) of LBW rats were significantly higher than those of control rats (Fig 2B).

## Renal glomerular morphology and aortic vascular smooth muscle staining

The kidneys of LBW rats and control rats exposed to a high-fat diet were excised and subjected to PAM staining. There was no difference in the number of glomeruli per field of view of the glomeruli (5.8±0.2, 5.9±0.5, 5.8±0.2 and 5.6±0.2 glomeruli/field of view in standard chow-fed NBW, standard chow-fed LBW, high-fat diet-exposed NBW and high-fat diet-exposed LBW, respectively, no significance), leading to no identification of glomerular hypertrophy, glomerular injury with mesangial proliferation, or matrix deposition (Fig 2C). Similarly, aortic smooth muscle was stained with anti-SMA antibody, but no difference was found in smooth muscle thickness among them (Fig 2D).

## Angiotensin II receptor expression

The expression levels of angiotensin II receptors (AT1 and AT2) in aortas and hearts were examined by real-time RT-PCR and western blotting. The expressions of mRNA and protein of AT1 in the aortas (53.4±2.0%, p<0.005 for mRNA and 83.5±1.7%, p<0.05 for protein, n = 8) and hearts (42.9±2.3%, p<0.001 for mRNA and 49.3±1.9%, p<0.001 for protein, n = 8) of LBW rats fed the standard diet were significantly lower than those in controls, respectively (Figs 3A, 3C, 4A and 4C). When a high-fat diet was provided, the expressions of mRNA and protein of AT1 in the aortas (23.4±1.1%, p<0.005 for mRNA and 56.1±1.3, p<0.005 for protein, n = 8) and hearts (27.9±4.5%, p<0.01 for mRNA and 31.3±1.4%, p<0.001 for protein, n = 8) of high-fat diet-exposed LBW rats were significantly lower than those in standard chow-fed LBW rats, respectively (Figs 3A, 3C, 4A and 4C). While the expression of AT2 in the aortas of LBW rats fed a standard diet was lower than that in the controls (67.1±5.7%, p<0.005 for mRNA and 83.3±3.3%, p<0.01 for protein, n = 8) (Fig 3B and 3D), the expression of AT2 in the hearts of LBW rats fed a standard diet was higher than that in the controls (186.2±17.0%,

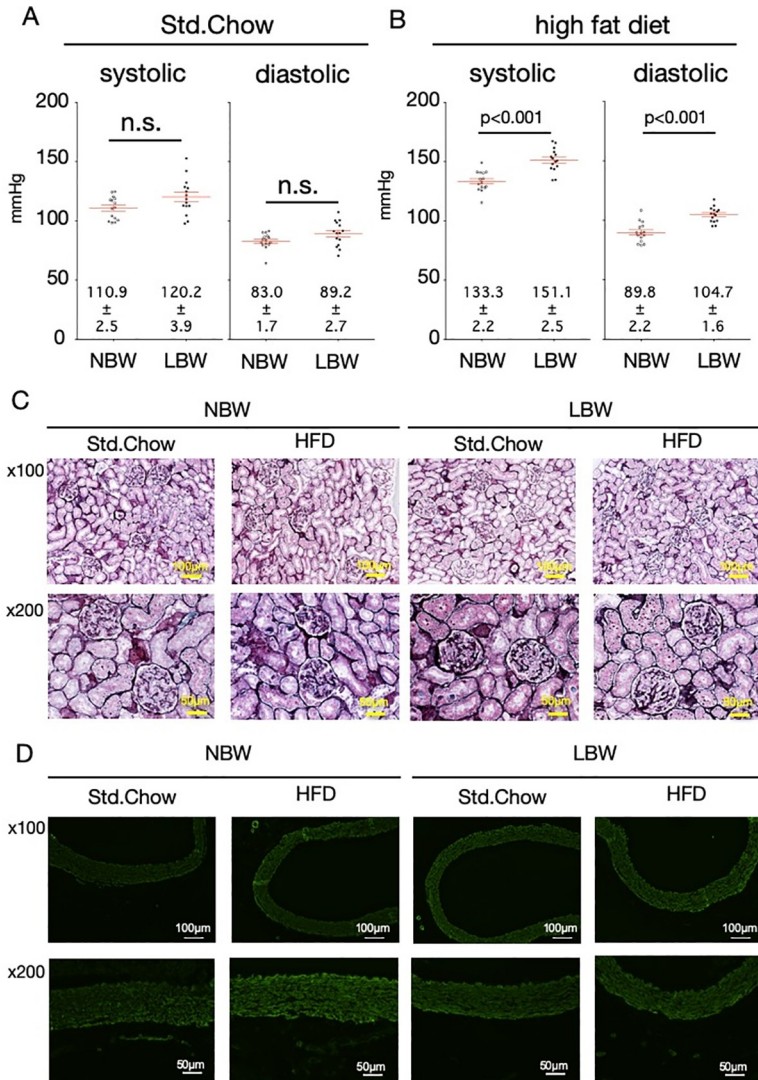

**Fig 2. Blood pressure and tissue observations of kidneys and aortas of LBW rats with high-fat diet exposure.** LBW rats were exposed to a high-fat diet (HFD) for 8 weeks. Blood pressure was measured by the tail-cuff method. (A) shows control rats (NBW) and LBW rats fed a standard chow, and (B) shows control rats and LBW rats exposed to a high-fat diet (n = 15). Statistical analysis was performed using unpaired T-test. Characteristic kidney PAM staining image and an aortic anti-SMA immunostaining image are shown (C and D).

p<0.01 for mRNA and 207.9±8.3%, p<0.001 for protein, n = 8) (Fig 4B and 4D). Exposure to a high-fat diet reduced their expression in the aorta (28.1±1.8%, p<0.005 for mRNA and 57.4 ±3.6%, p<0.005 for protein, n = 8) (Fig 3C and 3D). The protein expression of AT2 in the heart of high-fat diet-exposed LBW was significantly increased compared with that of standard chow-fed LBW (234.0±7.7%, p<0.05, n = 8) (Fig 4D).

## Blood hormone levels

Blood aldosterone and corticosterone levels in LBW rats fed a standard diet did not differ from those of control rats (Fig 5). When a high-fat diet was given to LBW rats, blood cortico-sterone levels increased significantly compared to control rats (684.3±23.8 ng/ml and 333.1 ±11.3 ng/ml, high-fat diet-exposed LBW and high-fat diet-exposed NBW, respectively,

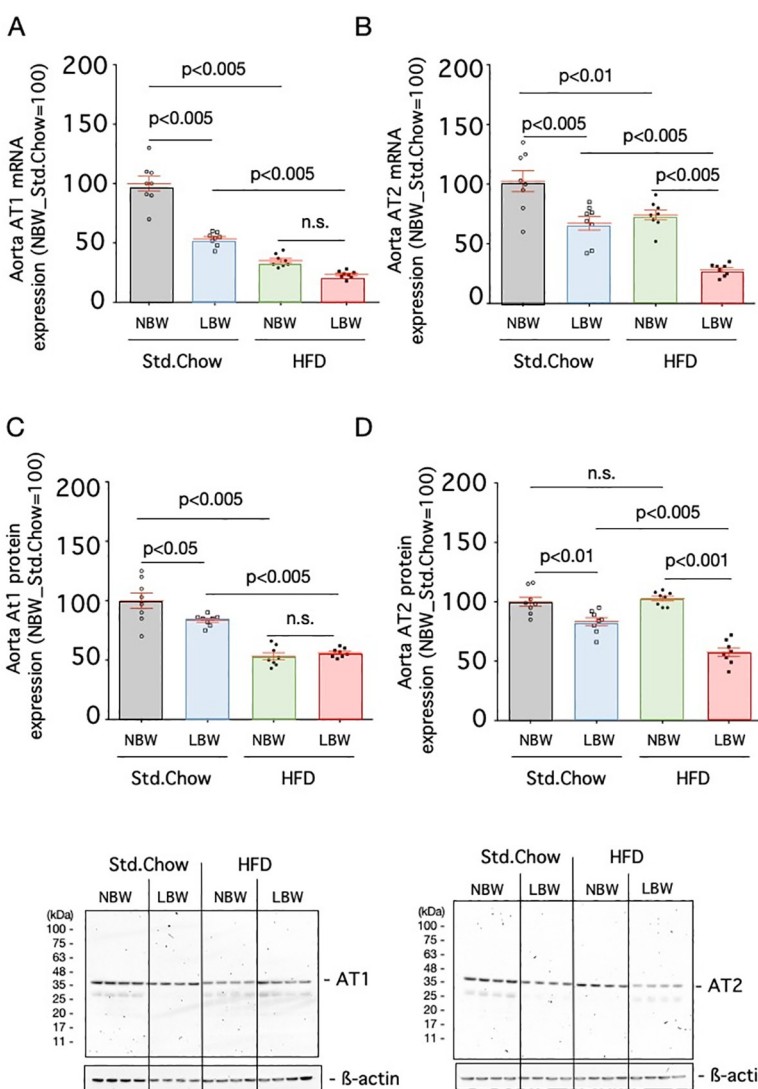

**Fig 3. Expression of angiotensin receptor in the aorta.** The mRNA expression levels of AT1 (A) and AT2 (B) and protein expression levels of AT1 (C) and AT2 (D) in the aortas of control rats (NBW) or LBW rats exposed to a standard chow (Std.Chow) or high-fat diet (HFD) were quantified. The mRNA expression level is a ratio obtained by correcting the ΔΔCT values of AT1 or AT2 with the ΔΔCT values of GAPDH and setting the value of Std.Chow-fed NBW to 100. The protein expression level is a ratio obtained by correcting the signal of AT1 or AT2 with the signal of ß-actin and setting the value of Std.Chow-fed NBW to 100. n = 8. Statistical analysis was performed using one-way ANOVA followed by Turkey's *post hoc* test for multiple comparisons.

$p < 0.005$, n = 7), but blood aldosterone levels did not differ between the two groups. Blood leptin levels did not differ between NBW (447.4±47.7 pg/ml) and LBW (455.0±48.7 pg/ml) rats fed a standard diet. The high-fat diet significantly increased leptin levels in NBW rats (3047 ±773.0 pg/ml, $p < 0.01$ vs. Std.Chow, n = 6) compared to standard chow-fed NBW rats, but not significantly in LBW (1193±359.0 pg/ml).

## Expression of miR-449a in the anterior pituitary

The expression of miR-449a in the pituitary glands of rats fed a standard diet was not different between control and LBW rats (Fig 6). When a high-fat diet was given, pituitary miR-449a was

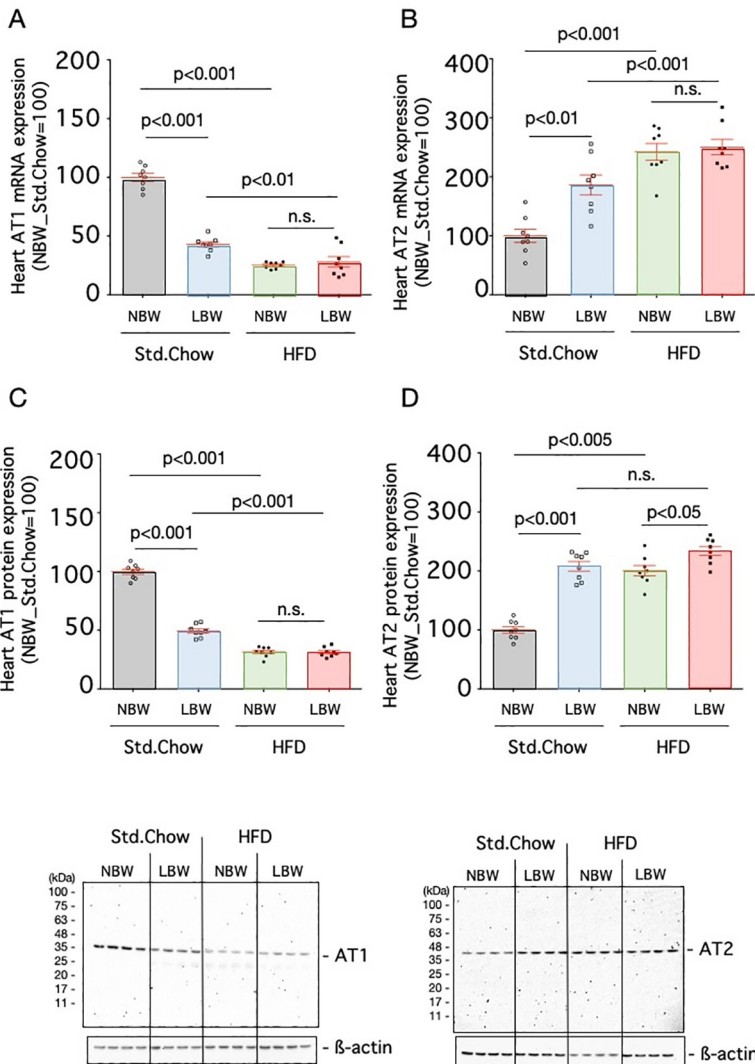

**Fig 4. Expression of angiotensin receptor in the heart.** The mRNA expression levels of AT1 (A) and AT2 (B) and protein expression levels of AT1 (C) and AT2 (D) in the hearts of control rats (NBW) or LBW rats exposed to a standard chow (Std.Chow) or high-fat diet (HFD) were quantified. The mRNA expression level is a ratio obtained by correcting the ΔΔCT values of AT1 or AT2 with the ΔΔCT values of GAPDH and setting the value of Std.Chow-fed NBW to 100. The protein expression level is a ratio obtained by correcting the signal of AT1 or AT2 with the signal of ß-actin and setting the value of Std.Chow-fed NBW to 100. n = 8. Statistical analysis was performed using one-way ANOVA followed by Turkey's *post hoc* test for multiple comparisons.

increased compared to the standard diet in controls (220.6±8.2% of control, p<0.001, n = 7); however, in LBW rats there was no increase in miR-449a with high-fat diet exposure (122.4 ±3.3% of control, p>0.05, n = 7).

## Administration of steroid synthesis inhibitors

Administration of metyrapone to 18-week-old LBW rats exposed to a high-fat diet significantly reduced the levels of corticosterone (70.8±14.7 ng/ml for metyrapone and 679.0±26.8 ng/ml for vehicle, p<0.001, n = 6) and aldosterone (51.0±5.9 ng/ml for metyrapone and 233.8 ±12.3 ng/ml for vehicle, p<0.001, n = 6), respectively, which were elevated by the high-fat diet (Fig 7A and 7B). Metyrapone significantly reduced the upregulation of blood pressure in LBW

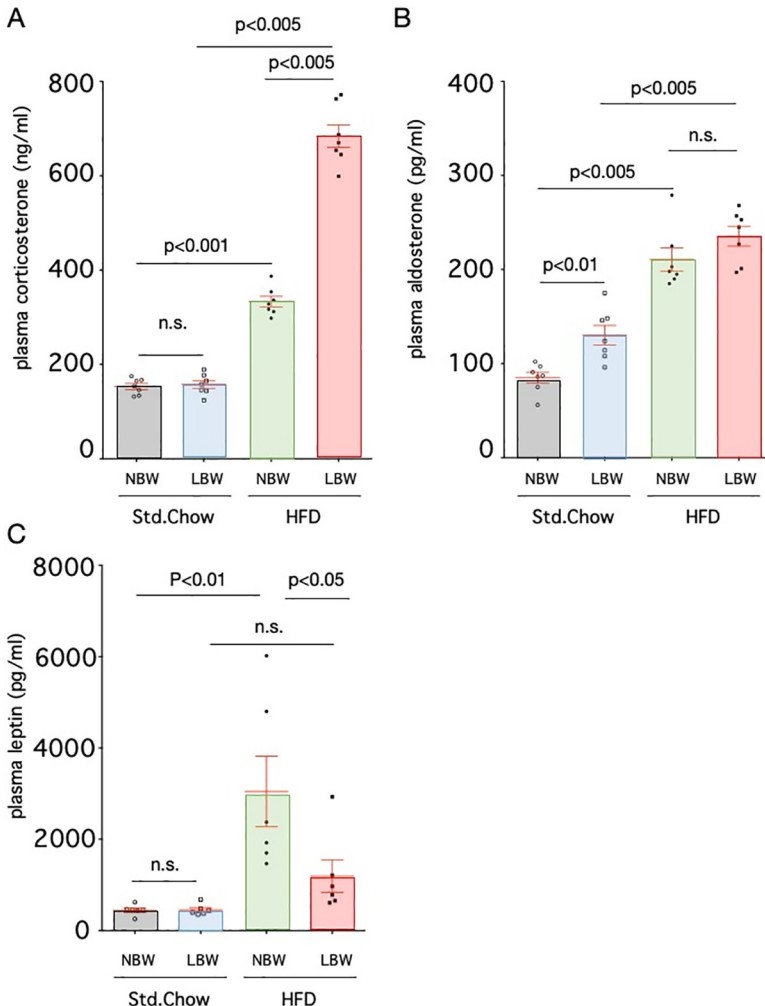

**Fig 5. Blood aldosterone, corticosterone and leptin levels.** Plasma concentrations of aldosterone (A), corticosterone (B) and leptin (C) of control rats (NBW) or LBW rats exposed to a standard chow or high-fat diet (HFD) were measured. Statistical analysis was performed using one-way ANOVA followed by Turkey's *post hoc* test for multiple comparisons.

rats to levels comparable with control rats (Fig 7C and 7D). Administration of metyrapone to control rats did not produce different results from vehicle administration.

## Discussion

In our rat model, elevated corticosterone levels (with renal sodium reabsorption) following exposure to a high-fat diet may be the main contributing factor for increased blood pressure in LBW rats. The expression of AT1, which is involved in increasing blood pressure by vasoconstriction, also decreased and AT2 expression, which is involved in decreasing blood pressure by vasodilation, was increased. These results suggest that the pattern of altered expression in the heart may be a compensatory change in blood pressure. Thus, all results, including the absence of pathological findings in the kidney and aorta as well as the elevation of blood aldosterone levels, indicated that renal and vascular diseases may not play a causative role in the development of hypertension in our full-term fetal low-carbohydrate and calorie-restricted rat model.

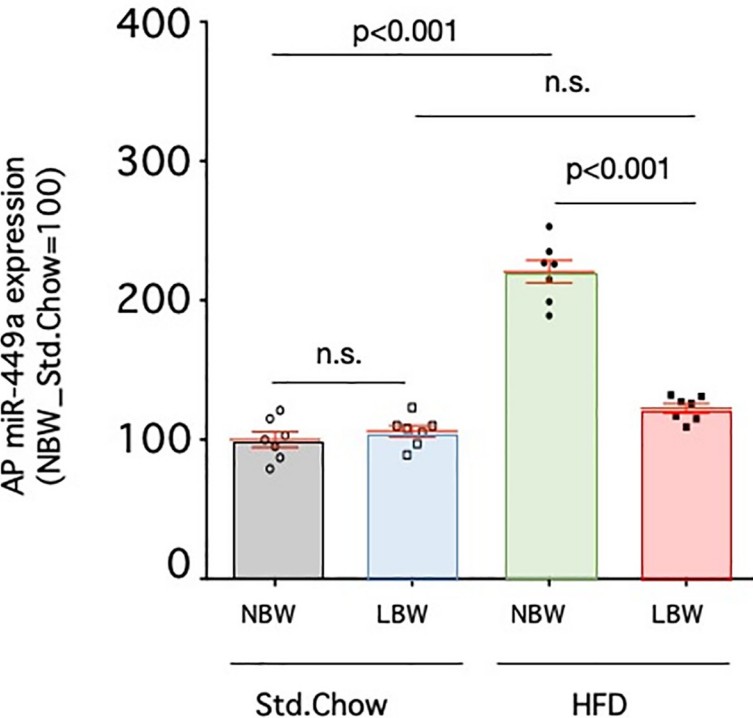

**Fig 6. Expression of miR-449a in the anterior pituitary.** Expression of miR-449a in the anterior pituitary of control rats (NBW) or LBW rats exposed to a standard chow (Std.Chow) or high-fat diet (HFD) were quantified. The miRNA expression level is a ratio obtained by correcting the ΔΔCT values of miR-449a with the ΔΔCT values of U6 and setting the value of Std.Chow-fed NBW to 100. n = 7. Statistical analysis was performed using one-way ANOVA followed by Turkey's *post hoc* test for multiple comparisons.

It has been reported that premature babies have low nephron numbers because the formation of renal glomeruli depends on the embryonic period. In our model, there was no difference in gestational age, as previously reported [34], and only birthweight was decreased. In fact, there was no difference in the weights of kidneys per body weight, and histological analysis of the kidney showed no differences in the number of glomeruli per visual field or in the morphological appearance. These findings suggest that the mechanism may potentially differ from the development of hypertension often seen in preterm infants. Furthermore, since the expression of AT1 was decreased in the hearts and aortas of high-fat diet-exposed LBW rats, these are considered to be compensatory changes. Although aldosterone levels in the blood increased with a high-fat diet compared to the standard chow, there was no difference between LBW and control rats. Moreover, there was no difference in heart rate. Although we did not investigate the functions of the autonomic nervous system, we consider that the involvement of the sympathetic nervous system may not be taken into account.

Protein intake is the key to increased blood pressure caused by fetal malnutrition. Maternal protein restriction during pregnancy has been reported to result in kidney abnormalities (decreased nephron number and glomerular morphology) and an increased renin-angiotensin system. Fujii *et al.* report that a 30% dietary restriction during pregnancy results in increased blood pressure in rat offspring, but branched-chain amino acid supplementation normalized it [40]. Allen and Zeman suggest that the progeny of protein-deficient rats show a permanent retardation in kidney development which is not reversed by increased postnatal nutrient intake [41]. On the contrary, postnatal high protein diet amplifies the renal damage induced

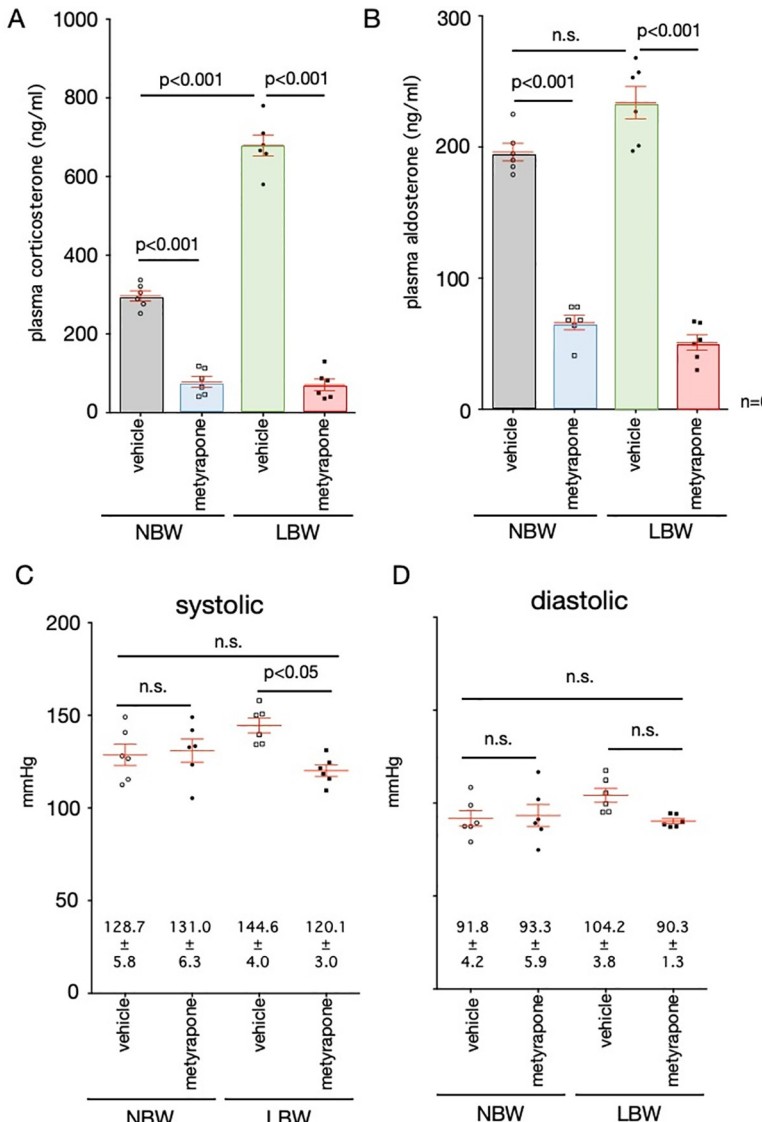

**Fig 7. Changes in blood pressure after administration of steroid synthesis inhibitors.** Blood corticosterone (A) and aldosterone (B) concentrations were measured in rats treated with metyrapone. The blood pressure of control rats (NBW) and LBW rats exposed to the high-fat diet (HFD) after the administration of metyrapone or vehicle was measured by the tail cuff method. The systolic pressure (C) and the diastolic pressure (D) of each rat are shown. n = 6. Statistical analysis was performed using one-way ANOVA followed by Turkey's *post hoc* test for multiple comparisons.

by fetal undernutrition [42]. Those results show that the amino acids consumed by the mother are important for the formation of the fetal and postnatal kidney. The effects of nutrients on nephron formation, especially the effects of placenta-mediated amino acids, need to be investigated. However, in our results, these abnormalities were not observed in a fetal low-carbohydrate calorie-restriction model. Elevated blood pressure in LBW rats exposed to a high-fat diet may not be related to renal nephron hypoplasia. Although detailed alteration of metabolites from low-carbohydrate and calorie-restriction exposure needs to be investigated in the near future, we have shown that the mechanisms of hypertension are different between protein-restriction and carbohydrate-restriction.

Singal *et al.* reviewed that atherosclerosis has a long preclinical stage in which pathological changes appear in the arteries of children and young adults before overt clinical manifestations of the disease appear [43]. Both fetal and childhood nutrition is important in this process and has been shown to affect the risk of lifelong CVD (e.g. breastfeeding is associated with long-term cardiovascular risk factors). It has been reported that fetal high-density lipoprotein cholesterol (HDL-C) and total cholesterol (TC) levels are lower in IUGR, and that the atherogenic index is increased in IUGR compared to healthy neonates [44]. Arteriosclerosis is an inflammatory disease, and its onset is known to involve excessive LDL-C [45] but may also involve a decrease in HDL-C. Because of the anti-inflammatory properties of HDL-C, a decrease in HDL-C during fetal development and an increase in atherogenic index may be involved in the future development of arteriosclerosis in IUGR children. Although we have not examined the blood lipid profile of this rat model, we did not observe thickening of vascular smooth muscle in the aorta. Furthermore, we found that the expression of AT2 was increased in the hearts of high-fat diet-exposed LBW rats compared to controls, but not in the aortas. AT2 is involved in the regulation of remodeling [30]. In the present study, the aortas of high-fat diet-exposed LBW rats may not have been damaged. However, future studies are needed to clarify that the details of the mechanisms underlying increase in AT2R expression and changes in the blood pressure control system that occur when AT2R expression is blocked. In fact, there is a report indicating that a major risk factor for the development of arteriosclerosis as well as hypertension is preterm birth, and the risk of that in full-term small for gestational age (SGA) children is not high [46]. Therefore, it is unlikely that atherosclerosis is not a principal cause for hypertension observed in our model.

Previous studies have reported that basal blood glucocorticoid levels in offspring are elevated by various factors, such as embryonic protein deficiency [47], maternal obesity [48, 49], prenatal stress [50, 51], and early life stress (including early weaning and maternal separation) [52–54]. The steroid synthesis inhibitor metyrapone normalized the abnormalities caused by glucocorticoids [55, 56]. Our embryonic low-carbohydrate, calorie restricted diet did not affect the basal blood corticosterone levels in offspring, but significantly increased after a high-fat diet exposure. The difference from the previous reports that affect the basal level of glucocorticoid is considered to be that embryonic hypoglycemic stress is relatively mild stress. The Predictive adaptive response hypothesis is thought to make a trade-off that alters the metabolic and endocrine systems to protect brain size [57]. It can be thought that the brain was protected by a trade-off in the mild stress of sugar restriction of dams. However, when a mismatch such as a high-fat diet occurs, it is possible that the abnormality cannot be withstood and the abnormality appears. Therefore, it is speculated that the basal blood corticosterone concentration was not affected and showed high level after a high-fat diet exposure. In the future, it is necessary to investigate the mechanism that causes embryonic programming of glucocorticoid feedback due to the difference in stress intensity.

Excess glucocorticoids often survive inactivation by 11β-hydroxysteroid dehydrogenase-2 in the kidney and bind to mineralocorticoid receptors. As a result, blood pressure increases due to sodium reabsorption and increased fluid volume. Ingestion of a high-fat diet or obesity results in metabolic stress, leading to an increase in blood glucocorticoids [58, 59]. Excessive and prolonged glucocorticoid exposure is associated with a high cardiovascular and metabolic burden. Chronic hypercortisolemia causes more persistent visceral fat accumulation than high-fat diet-induced obesity [60]. Thus, prolonged exposure to high-fat diets can generate a negative loop of metabolic disease. We have previously reported that LBW rats show long-term high blood corticosterone after restraint stress, and that corticotropin releasing factor receptor (CRF-R1) fails to downregulate without inducing miR-449a expression in the pituitary gland [17, 18]. Since pituitary miR-449a is induced by glucocorticoids and downregulates

CRF-R1 expression, miR-449a is thought to be one of the mechanisms of negative feedback of the HPA-axis by glucocorticoids. In the present study, we showed that blood corticosterone levels in high-fat diet-exposed LBW rats were higher than that in control rats. Pituitary miR-449a is induced in high-fat diet-exposed control rats compared to standard diet-fed control rats, but miR-449a did not increase in high-fat diet-exposed LBW rats, and the mechanism is unknown. In addition, we have shown that administration of steroid synthesis inhibitors to high-fat diet-exposed LBW rats normalized their blood pressure. These results suggest that LBW rats may have abnormal pituitary glucocorticoid feedback of the HPA-axis due to metabolic stress induced by high-fat diet exposure, and blood pressure may have increased due to the increased blood corticosterone concentration.

In conclusion, we found that the mechanism of elevated blood pressure in high-fat diet-exposed LBW rats is associated with abnormal glucocorticoid negative feedback due to increased miR-449a expression in the pituitary. There are various causes of the risk of developing hypertension due to LBW, but it is necessary to develop a treatment strategy for hypertension for each cause of LBW. One of the mechanisms of hypertension in high-fat diet-exposed LBW is suggested to be related to elevated blood corticosterone levels in our model rats. We think that the mismatch between the acquired constitution by undernutrition and the environment induced by a high-fat diet after growth increases the risk of developing metabolic disease. Our results imply that LBW children resulting from low-carbohydrate and calorie-restricted diets in mothers should be carefully followed in terms of evaluating their blood cortisol levels.

## Supporting information

**S1 Table.**
(DOCX)

**S2 Table.**
(XLSX)

**S1 Raw images.**
(PDF)

## Author Contributions

**Conceptualization:** Takahiro Nemoto.

**Data curation:** Takahiro Nemoto, Takashi Nakakura.

**Formal analysis:** Takahiro Nemoto, Takashi Nakakura.

**Funding acquisition:** Takahiro Nemoto.

**Investigation:** Takahiro Nemoto.

**Methodology:** Takahiro Nemoto.

**Resources:** Takahiro Nemoto.

**Supervision:** Yoshihiko Kakinuma.

**Validation:** Yoshihiko Kakinuma.

**Writing – original draft:** Takahiro Nemoto.

**Writing – review & editing:** Takashi Nakakura, Yoshihiko Kakinuma.

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
