## [Decision Letter · Decision Letter 0]

30 Jun 2020

PONE-D-20-16514

Elevated blood pressure in high-fat diet-exposed low birthweight rat offspring is most likely caused by elevated glucocorticoid levels

PLOS ONE

Dear Dr. Nemoto,

Thank you for submitting your manuscript to PLOS ONE. After careful consideration, we feel that it has merit but does not fully meet PLOS ONE’s publication criteria as it currently stands. Therefore, we invite you to submit a revised version of the manuscript that addresses the points raised during the review process.

We look forward to receiving your revised manuscript.

Kind regards,

Michael Bader

Academic Editor

PLOS ONE

Journal Requirements:

'The authors have nothing to disclose.'

a. Please complete your Competing Interests statement to state any Competing Interests.

If you have no competing interests, please state "The authors have declared that no competing interests exist.", as detailed online in our guide for authors at http://journals.plos.org/plosone/s/submit-now

Reviewers' comments:

Reviewer's Responses to Questions

**Comments to the Author**

1. Is the manuscript technically sound, and do the data support the conclusions?

Reviewer #1: No

Reviewer #2: Yes

2. Has the statistical analysis been performed appropriately and rigorously? 

Reviewer #1: No

Reviewer #2: Yes

3. Have the authors made all data underlying the findings in their manuscript fully available?

Reviewer #1: No

Reviewer #2: Yes

4. Is the manuscript presented in an intelligible fashion and written in standard English?

Reviewer #1: Yes

Reviewer #2: Yes

5. Review Comments to the Author

Reviewer #1: This study investigated the mechanisms by which LBW full term causes increased BP.

Comments to the authors:

Abstract:

The abstract states that the mechanisms linking BW and BP are unknown for full-term LBW. Yet, numerous studies using different experimental models of LBW that do not deliver preterm including models of DOHaD induced via placental insufficiency, fetal exposure to GCs or maternal nutrient restriction (Langley-Evans, Baum, Wlodek, Vehaskari, Zambrano, Alexander, Rose, and Woods to name just a few) report important roles for the RAS, SNS, ET, GCs and ROS. Thus, this statement is misleading.

What is the rational for use of metyrapone, for study of miR-449a?

Overall, the abstract could provide better rationale for the study.

Cite the degree of nutrient restriction; source of carbohydrates in the prenatal diet. What is the percent fat and source of fat in the high fat diet?

Introduction:

Cites reviews not actual research papers for background. If citing reviews, then state as such.

Methods:

Details related to diets are absent, grossly limited and make it hard to compare this study to other studies. It is well-established that subtle differences in micro- and macro-nutrients during development can lead to significant differences in phenotypic outcomes.

Use of tail cuff for measurement of BP is not the gold standard and is well-established to demonstrate stress-induced BP. Why was BP not confirmed using a direct method?

Use of acid maceration is not the gold standard for detecting differences in nephron number (Bertram JF. Kid Internal. 2001).

How many dams provided offspring? Was only one offspring per sex per litter per dietary exposure utilized for each study parameter? This is critical to ensure that studies are not reporting litter effects versus a programming effect.

Antibodies for detection of Ang II receptors lack specificity (Herrera et al. Hypertension 2013;61:253.)

Why were Ang II components not studied within the kidney?

Results:

The authors state this is a non-preterm model. Did gestational length differ? Why was this not tested?

For body weight, provide ages at time of data presentation; add this to the text of the results.

N=50 and N=59? Break values into those LBW remaining on a normal diet versus those switched to the high fat diet; ditto for NBW. Thus 4 groups with smaller N.

Yet, why are only 15 and 15 rats per LBW and NBW on high and normal diet used, only a total of 30 for LBW and 30 for NBW? What happened to the other 20 and 29 rats?

Blood pressure: state age at time of measurement in the text of the results?

Related studies not considered that show that blockade of GC synthesis in models of DOHaD:

Murphy et al AJP: Endo Metab 2016;312:E98, ELS and obesity, role of GCs using metyrapone

Reviewer #2: In this manuscript, the authors present the outcome of experiments designed to show that LBW rats (generated from fetal low-carbohydrate and calorie-restricted maternal diet) had significantly higher blood pressure than controls when exposed to a high-fat diet as newborns. No observed changes in nephron number or size were reported; however, aortic and cardiacangiotensin II receptor expression was altered. Blood hormone analysis shows no changes in aldosterone, but corticosterone levels were elevated in high-fat diet fed newborns, with metyrapone treatment inhibiting the elevation in blood pressure. The authors further report that that high-fat diet exposure causes impairment of the pituitary glucocorticoid feedback via miR-449a. The study is an extension to their previously published work utilizing the same fetal programming model which showed a heightened glucocorticoid response in response to restraint stress, suggesting dysfunctional glucocorticoid-mediated feedback system in the pituitary gland. The authors have also previously shown that pituitary miR-449a is involved in the negative feedback of glucocorticoids, and restraint stress-induced LBW rats have impaired induction of miR-449a expression in the anterior pituitary gland. The current study shows that high fat diet, similar to restraint stress have impaired miR-449a expression, which would not be surprising, considering that a reduction in miR-449 is pre-programmed by the LBW as they have previously reported. I do have some concerns and these are outlined below.

1. Number of animals used in the study should be clearly described. How many female animals were used for breeding, how many offsprings were used from each litter and how were they assigned for each study/experimental analysis?

2. It is surprising that the body weight of LBW rats on standard diet was not statistically different from NBW rats on standard diet. Can the authors provide an explanation, as this is different from previous reported observations in similar programming models.

3. What is the interpretation and significance of the finding that AT2R is elevated in both LBW and NBW following high fat diet?

4. A number of animal models/studies (high fat diet, caloric restriction, hypoxia, maternal stress, etc.) would suggest and have shown elevated levels of glucocorticoids postnatally and is the key mediator of in utero programming and has been shown to be elevated in LBW offsprings; however it appears that low carb/caloric restriction model used by the authors does not show differences in blood cort levels between LBW and NBW animals. Please elaborate in discussion as to the difference in findings, and it’s relevance; however, it is intriguing that the authors are showing elevated aldosterone levels in LBW. Similarly, it is interesting that the low carb/caloric restriction model does not show increased BP in LBW compared to NBW; however, it does show an increase following restraint stress (previously published) and now HFD. The authors should highlight the significance of these observations relevant to current literature and the model used may be different than other models used to study DoHAD, and how the current model is relevant to human studies/patients..

5. Although the authors demonstrate that glucocorticoids is involved in increased BP induced by postnatal high-fat diet in LBW newborns, these findings are not new, as glucocorticoids have been widely reported to be involved in programming of hypertension in adult animal studies..

6. The authors report the absence of pathological findings in the kidney and aorta as well as the elevation of blood aldosterone levels, indicated that renal and vascular diseases may not play a causative role in the development of hypertension in fetal low-carbohydrate and calorie-restricted rats. A potential mechanism for increased BP should be discussed in light of the negative findings in the renal and vasculature, considering the findings are contrary to what is observed in premature babies and other models of programming.

6. PLOS authors have the option to publish the peer review history of their article (what does this mean?). If published, this will include your full peer review and any attached files.

Reviewer #1: No

Reviewer #2: No

---

## [Author Response · Author response to Decision Letter 0]

23 Jul 2020

Reviewer #1: This study investigated the mechanisms by which LBW full term causes increased BP.

Comments to the authors:

Abstract: The abstract states that the mechanisms linking BW and BP are unknown for full-term LBW. Yet, numerous studies using different experimental models of LBW that do not deliver preterm including models of DOHaD induced via placental insufficiency, fetal exposure to GCs or maternal nutrient restriction (Langley-Evans, Baum, Wlodek, Vehaskari, Zambrano, Alexander, Rose, and Woods to name just a few) report important roles for the RAS, SNS, ET, GCs and ROS. Thus, this statement is misleading.

 >According to the reviewer’s comment, we have revised the abstract text.

What is the rational for use of metyrapone, for study of miR-449a?

 >We reported in a previous paper that glucocorticoids increased miR-449a expression. Therefore, we believe that the increased expression of miR-449a suppresses the expression of Crhr1 and causes negative feedback of glucocorticoid. The elevated blood pressure in our model rat is due to high levels of corticosterone, and miR-449a does not directly affect suppression of blood pressure. Thus, in this paper, there is no relationship between expression of miR-449a and metyrapone.

Overall, the abstract could provide better rationale for the study.

 >According to the reviewer’s comment, we have revised the abstract text.

Cite the degree of nutrient restriction; source of carbohydrates in the prenatal diet. What is the percent fat and source of fat in the high fat diet?

 >According to the reviewer’s comment, we've added details on calorie-restricted (60% of standard chow) and high-fat diets (lard-based 45kcal% fat).

Introduction: Cites reviews not actual research papers for background. If citing reviews, then state as such.

 >According to the reviewers' comments, we removed some of the reviews cited and stated that the remaining review.

Methods: Details related to diets are absent, grossly limited and make it hard to compare this study to other studies. It is well-established that subtle differences in micro- and macro-nutrients during development can lead to significant differences in phenotypic outcomes.

 >We showed a schematic of the diets fed to the dams in Fig 1A, but provided the details as a supplementary table 1.

Use of tail cuff for measurement of BP is not the gold standard and is well-established to demonstrate stress-induced BP. Why was BP not confirmed using a direct method?

 >As the reviewer’s comment, we understand that it is better to use a telemetry system. Unfortunately, however, our laboratory did not have a telemetry system, and we were unable to perform measurements on rats in long-term care. We prepared the environment for measurement of blood pressure and showed the data with little variation by performing measurement 10 times or more in the same rat.

Use of acid maceration is not the gold standard for detecting differences in nephron number (Bertram JF. Kid Internal. 2001).

 >As the reviewer’s comment, we understand that acid maceration can provide a quick and reliable estimate of nephron counts throughout the kidney. Since we also observed the nephron structure, we stained with the PAM method. Since no abnormalities were found in the number and structure of nephrons in this study, only the results of PAM staining are shown.

We have shown that high salt exposure in our LBW rats cause polyuria and elevated blood pressure. In a follow-up report that will be compiled in the near future, we will explain the results of kidney analysis by comparing them with the high-fat diet exposure.

How many dams provided offspring? Was only one offspring per sex per litter per dietary exposure utilized for each study parameter? This is critical to ensure that studies are not reporting litter effects versus a programming effect.

 >In this study, 12 to 20 rat pups were obtained from each of 10 dams. We excluded pups born with a body weight of more than 6.0 g, which is the average-2SD body weight of the offspring of normal dams. No surrogate mother was used, and 10 rat pups were left at random and raised under the birth mother rat. After weaning, the rats of different litters were mixed and randomly divided into a high fat diet exposure group and a standard diet feeding group for use in the experiment. According to the reviewer’s comment, we've added details in method section.

Antibodies for detection of Ang II receptors lack specificity (Herrera et al. Hypertension 2013;61:253.)

 >The antibody used in this manuscript may lack specificity, as the reviewer commented. However, since it showed the same pattern as mRNA expression and a different expression pattern from AT1R, we determined that we could obtain or detect the alterations in ATR2 expression.

Why were Ang II components not studied within the kidney?

 >We have found that high fat diet-exposed LBW decreased AT1R mRNA expression and increased AT2R mRNA expression in the kidney as shown in the heart. However, when LBW was exposed to a high salt diet, renal AT1R mRNA expression was significantly increased compared to control rats exposed to a high fat diet. As mentioned above, polyuria and elevated blood pressure are seen in high salt diet-exposed LBW. Unlike the high-fat diet exposure, it was suggested that the kidney is involved in the high salt exposure, so we are preparing a follow-up report. The expressions of AT1R and AT2R in the kidney will be reported in a follow-up report comparing high fat and high salt diet exposures.

Results: The authors state this is a non-preterm model. Did gestational length differ? Why was this not tested?

 >As previously reported by Sci Rep (2020), there was no difference in gestational age. We have added this to the results.

For body weight, provide ages at time of data presentation; add this to the text of the results.

 >According to the reviewer’s comment, we have added the age of the rats to the text.

N=50 and N=59? Break values into those LBW remaining on a normal diet versus those switched to the high fat diet; ditto for NBW. Thus 4 groups with smaller N.

Yet, why are only 15 and 15 rats per LBW and NBW on high and normal diet used, only a total of 30 for LBW and 30 for NBW? What happened to the other 20 and 29 rats?

 >In this study, we used 15 rats of each group to measure blood pressure, 8 rats of each group to analyze blood hormones and mRNA expression, and 6 rats of each group to measure blood pressure after administration of metyrapone (We used the same rat for blood pressure measurement and hormone concentration measurement after administration of metyrapone.) The graph of birth weight in Fig. 1B was revised because the data of birthweight of the rat used for the administration of metyrapone were missing. In addition, we have revised the total number of animals in Fig. 1D and Fig. 1E.

Blood pressure: state age at time of measurement in the text of the results?

 >According to the reviewer’s comment, we have added the age of the rats to the text.

Related studies not considered that show that blockade of GC synthesis in models of DOHaD: Murphy et al AJP: Endo Metab 2016;312:E98, ELS and obesity, role of GCs using metyrapone.

 >Thank you for presenting the paper. According to the reviewer's comments, we have added the following text to the discussion and cited the paper in this section. 

Previous studies have reported that basal blood glucocorticoid levels in offspring are elevated by various factors, such as preterm birth, embryonic protein deficiency, prenatal stress, and early postnatal stress (early weaning and maternal separation). The steroid synthesis inhibitor metyrapone normalized the abnormalities caused by glucocorticoids. Our embryonic low-carbohydrate, calorie restricted diet did not affect the basal blood corticosterone levels in offspring, but significantly increased after a high-fat diet exposure. The difference from the previous reports that affect the basal level of glucocorticoid is considered to be that embryonic hypoglycemic stress is relatively mild stress. “The Predictive adaptive response hypothesis” is thought to make a trade-off that alters the metabolic and endocrine systems to protect brain size. It can be thought that the brain was protected by a trade-off in the mild stress of sugar restriction of dams. However, when a mismatch such as a high-fat diet occurs, it is possible that the abnormality cannot be withstood and the abnormality appears. Therefore, it is speculated that the basal blood corticosterone concentration was not affected and showed a high level after a high-fat diet. In the future, it is necessary to investigate the mechanism that causes embryonic programming of glucocorticoid feedback due to the difference in stress intensity.

Reviewer #2: In this manuscript, the authors present the outcome of experiments designed to show that LBW rats (generated from fetal low-carbohydrate and calorie-restricted maternal diet) had significantly higher blood pressure than controls when exposed to a high-fat diet as newborns. No observed changes in nephron number or size were reported; however, aortic and cardiac angiotensin II receptor expression was altered. Blood hormone analysis shows no changes in aldosterone, but corticosterone levels were elevated in high-fat diet fed newborns, with metyrapone treatment inhibiting the elevation in blood pressure. The authors further report that that high-fat diet exposure causes impairment of the pituitary glucocorticoid feedback via miR-449a. The study is an extension to their previously published work utilizing the same fetal programming model which showed a heightened glucocorticoid response in response to restraint stress, suggesting dysfunctional glucocorticoid-mediated feedback system in the pituitary gland. The authors have also previously shown that pituitary miR-449a is involved in the negative feedback of glucocorticoids, and restraint stress-induced LBW rats have impaired induction of miR-449a expression in the anterior pituitary gland. The current study shows that high fat diet, similar to restraint stress have impaired miR-449a expression, which would not be surprising, considering that a reduction in miR-449 is pre-programmed by the LBW as they have previously reported. I do have some concerns and these are outlined below.

1. Number of animals used in the study should be clearly described. How many female animals were used for breeding, how many offsprings were used from each litter and how were they assigned for each study/experimental analysis?

 >In this study, 12 to 20 rat pups were obtained from 10 dams of each group. We excluded pups born with a body weight of more than 6.0 g, which is the average-2SD body weight of the offspring of normal dams. No surrogate mother was used, and 10 rat pups were left at random and raised under the birth mother rat. After weaning, the rats of different litters were mixed and randomly divided into a high fat diet exposure group and a standard diet feeding group for use in the experiment. 

In this study, we used 15 rats of each group to measure blood pressure, 8 rats of each group to analyze blood hormones and mRNA expression, and 6 rats of each group to measure blood pressure after administration of metyrapone (We used the same rat for blood pressure measurement and hormone concentration measurement after administration of metyrapone.) The graph of birth weight in Fig. 1B was revised because the data of birthweight of the rat used for the administration of metyrapone were missing. In addition, we have revised the total number of animals in Fig. 1D and Fig. 1E.

2. It is surprising that the body weight of LBW rats on standard diet was not statistically different from NBW rats on standard diet. Can the authors provide an explanation, as this is different from previous reported observations in similar programming models.

 >Approximately 90% of low birth weight rats catch-up by the weaning period, and have an average weight similar to that of control rats. In the previous report on Scientific Reports (2020), we collected and analyzed rats that did not catch-up by the weaning period, but this time we analyzed collectively with or without catch-up. In our preliminary experiment, the blood pressure of LBW with high fat diet exposure was higher than that of control rats with high fat diet, with or without catch-up growth. Therefore, in this paper, we presented the results that include the results of the catch-up rats. (Because only 10% of the rats do not catch-up, it is difficult to prepare a number for use in various experiments.)

3. What is the interpretation and significance of the finding that AT2R is elevated in both LBW and NBW following high fat diet?

 >Since AT2R has vasodilatory and hypotensive effects, it is considered that the increased expression of AT2R in high-fat diet-exposed rats is the compensatory effect. We think that the details of the underlying mechanisms require further study, by which increase in AT2R expression and changes in the blood pressure control system that occur when AT2R expression is blocked. We have added these points to the discussion section.

4. A number of animal models/studies (high fat diet, caloric restriction, hypoxia, maternal stress, etc.) would suggest and have shown elevated levels of glucocorticoids postnatally and is the key mediator of in utero programming and has been shown to be elevated in LBW offspring; however, it appears that low carb/caloric restriction model used by the authors does not show differences in blood cort levels between LBW and NBW animals. Please elaborate in discussion as to the difference in findings, and it’s relevance; however, it is intriguing that the authors are showing elevated aldosterone levels in LBW. Similarly, it is interesting that the low carb/caloric restriction model does not show increased BP in LBW compared to NBW; however, it does show an increase following restraint stress (previously published) and now HFD. The authors should highlight the significance of these observations relevant to current literature and the model used may be different than other models used to study DoHAD, and how the current model is relevant to human studies/patients.

 >According to the reviewer's comments, we have added the following to the discussion. 

Previous studies have reported that basal blood glucocorticoid levels in offspring are elevated by various factors, such as preterm birth, embryonic protein deficiency, prenatal stress, and early postnatal stress (early weaning and maternal separation). The steroid synthesis inhibitor metyrapone normalized the abnormalities caused by glucocorticoids. Our embryonic low-carbohydrate, calorie restricted diet did not affect the basal blood corticosterone levels in offspring, but significantly increased after a high-fat diet exposure. The difference from the previous reports that affect the basal level of glucocorticoid is considered to be that embryonic hypoglycemic stress is relatively mild stress. The Predictive adaptive response hypothesis is thought to make a trade-off that alters the metabolic and endocrine systems to protect brain size. It can be thought that the brain was protected by a trade-off in the mild stress of sugar restriction of dams. However, when a mismatch such as a high-fat diet occurs, it is possible that the abnormality cannot be withstood and the abnormality appears. Therefore, it is speculated that the basal blood corticosterone concentration was not affected and showed a high level after a high-fat diet. In the future, it is necessary to investigate the mechanism that causes embryonic programming of glucocorticoid feedback due to the difference in stress intensity.

5. Although the authors demonstrate that glucocorticoids is involved in increased BP induced by postnatal high-fat diet in LBW newborns, these findings are not new, as glucocorticoids have been widely reported to be involved in programming of hypertension in adult animal studies.

 >According to the reviewer’s comment, we rewritten as finding that the mechanism of elevated blood pressure in high-fat diet-exposed low-birth-weight rats is associated with abnormal glucocorticoid feeder back due to increased miR-449a expression in the pituitary. We also changed the title to “Elevated blood pressure in high-fat diet-exposed low birthweight rat offspring is most likely caused by elevated glucocorticoid levels due to abnormal pituitary negative feedback”

6. The authors report the absence of pathological findings in the kidney and aorta as well as the elevation of blood aldosterone levels, indicated that renal and vascular diseases may not play a causative role in the development of hypertension in fetal low-carbohydrate and calorie-restricted rats. A potential mechanism for increased BP should be discussed in light of the negative findings in the renal and vasculature, considering the findings are contrary to what is observed in premature babies and other models of programming.

 >We understand that low birth weight infants have different mechanisms of increasing blood pressure in preterm and term infants. Our rat model is a term model because the gestational age was not shortened. As mentioned in the second paragraph of the discussion, we interpret that the glomerular number was not affected in our model because the number of glomeruli is strongly influenced by gestational age. Regarding the effects of nutrients in the uterus, we are planning an experiment because it is necessary to examine in detail how the carbohydrate restriction of the mother rat affected the amino acid composition of placental blood. We have added to the third paragraph of the discussion. It has been shown that the effects of glucocorticoids are affected not only during the embryonic period but also during postnatal stress. In the fourth comment from the reviewer, we added some thought to the involvement of glucocorticoids in blood pressure control in LBW rats.

---

## [Decision Letter · Decision Letter 1]

7 Aug 2020

PONE-D-20-16514R1

Elevated blood pressure in high-fat diet-exposed low birthweight rat offspring is most likely caused by elevated glucocorticoid levels due to abnormal pituitary negative feedback

PLOS ONE

Dear Dr. Nemoto,

Thank you for submitting your manuscript to PLOS ONE. After careful consideration, we feel that it has merit but does not fully meet PLOS ONE’s publication criteria as it currently stands. Therefore, we invite you to submit a revised version of the manuscript that addresses the points still raised by reviewer 1.

We look forward to receiving your revised manuscript.

Kind regards,

Michael Bader

Academic Editor

PLOS ONE

Reviewers' comments:

Reviewer's Responses to Questions

**Comments to the Author**

1. If the authors have adequately addressed your comments raised in a previous round of review and you feel that this manuscript is now acceptable for publication, you may indicate that here to bypass the “Comments to the Author” section, enter your conflict of interest statement in the “Confidential to Editor” section, and submit your "Accept" recommendation.

Reviewer #1: (No Response)

2. Is the manuscript technically sound, and do the data support the conclusions?

Reviewer #1: No

3. Has the statistical analysis been performed appropriately and rigorously? 

Reviewer #1: Yes

4. Have the authors made all data underlying the findings in their manuscript fully available?

Reviewer #1: No

5. Is the manuscript presented in an intelligible fashion and written in standard English?

Reviewer #1: Yes

6. Review Comments to the Author

Reviewer #1: The following concerns remain:

Weight at birth equals ~58 per group. This data should be presented as an average weight at birth per litter for the pups utilized for additional studies (excludes those culled).

Did average birth weight per litter for all viable pups (culled and utilized for additional studies) differ from average birth weight per litter for pups used for additional studies (excludes culled).

Methods needs to describe on page 6 the number of pups separated at 5 weeks of age that are in each experimental and dietary group.

A major concern is that after weaning, pups from all litters were mixed. Thus, can the authors ensure that only one pup per sex per litter was present in each study group versus multiple pups from the same litter included in the same dietary group. This is imperative in order to ensure a “programming” effect versus a “litter” effect.

Conclusions included on Page 11 within the first paragraph of the discussion that report an absence of pathological changes in the kidney and aorta (a non-resistance vessel) indicates a lack of involvement of renal and vascular disease as a component in the etiology of hypertension in this model is quite a stretch.

Conclusions that the mechanisms in this model differ from preterm infants is not unusual as this is not a model of preterm birth (line 287-288).

7. PLOS authors have the option to publish the peer review history of their article (what does this mean?). If published, this will include your full peer review and any attached files.

Reviewer #1: No

---

## [Author Response · Author response to Decision Letter 1]

11 Aug 2020

Reviewer #1: The following concerns remain:

Weight at birth equals ~58 per group. This data should be presented as an average weight at birth per litter for the pups utilized for additional studies (excludes those culled).

Did average birth weight per litter for all viable pups (culled and utilized for additional studies) differ from average birth weight per litter for pups used for additional studies (excludes culled).

 >The weights of all offspring used in the experiment are shown in Supplementary table 2. Female rats were not used in the experiment, but there was no difference between the average weight including female rats and the average weight of male rats (used in the experiment).

Methods needs to describe on page 6 the number of pups separated at 5 weeks of age that are in each experimental and dietary group.

 >According to the reviewer’s comment, the number of rats was revised. We used control and LC offspring (46 rats each) divided into two groups (23 rats each); a standard diet and a high-fat diet.

A major concern is that after weaning, pups from all litters were mixed. Thus, can the authors ensure that only one pup per sex per litter was present in each study group versus multiple pups from the same litter included in the same dietary group. This is imperative in order to ensure a “programming” effect versus a “litter” effect.

 >We randomly mixed rat pups and used them in the experiment, but divided them so that each group contained one offspring from 10 dams. For example, the systolic blood pressure shows a mean blood pressure of 15 rats with 1 offspring from LC dam# 1- #5 plus 2 offspring from LC dam#6 - #10. The systolic blood pressure of a high fat diet exposed-offspring from LC dam #6 was 150.1 mmHg in one and 165.8 mmHg in the other. The high fat diet exposed-offspring from LC dam #7 was 146.1 mmHg on one and 154.0 mmHg on the other. They were higher than those of high fat diet exposed-control offspring. In other words, blood pressure does not increase only in the group born from a particular dam, and low birth weight rat pups due to fetal malnutrition tend to have increased blood pressure when exposed to a high-fat diet after birth. All data (including the dam ID) is available on figshare.com (https://doi.org/10.6084/m9.figshare.12644165). 

Conclusions included on Page 11 within the first paragraph of the discussion that report an absence of pathological changes in the kidney and aorta (a non-resistance vessel) indicates a lack of involvement of renal and vascular disease as a component in the etiology of hypertension in this model is quite a stretch.

Conclusions that the mechanisms in this model differ from preterm infants is not unusual as this is not a model of preterm birth (line 287-288).

 >According to reviewer’s comment, we revised sentence in discussion section to “renal and vascular diseases may not play a causative role in the development of hypertension in our full-term fetal low-carbohydrate and calorie-restricted rat model.” (Changed to line 282-283)

---

## [Editor Report · Decision Letter 2]

13 Aug 2020

Elevated blood pressure in high-fat diet-exposed low birthweight rat offspring is most likely caused by elevated glucocorticoid levels due to abnormal pituitary negative feedback

PONE-D-20-16514R2

Dear Dr. Nemoto,

We’re pleased to inform you that your manuscript has been judged scientifically suitable for publication and will be formally accepted for publication once it meets all outstanding technical requirements.

Kind regards,

Michael Bader

Academic Editor

PLOS ONE
---

## [Editor Report · Acceptance letter]

17 Aug 2020

PONE-D-20-16514R2 

Elevated blood pressure in high-fat diet-exposed low birthweight rat offspring is most likely caused by elevated glucocorticoid levels due to abnormal pituitary negative feedback 

Dear Dr. Nemoto:

I'm pleased to inform you that your manuscript has been deemed suitable for publication in PLOS ONE. Congratulations! Your manuscript is now with our production department. 

Kind regards, 

on behalf of

Prof. Michael Bader 

Academic Editor

PLOS ONE